# Multilayer Model Predictive Control for a Voltage Mode Digital Power Amplifier

**Xinwei Wei** [1,2]**, Xiaonan Zhu** [1,2]**, Wenyuan Zhang** [1,2]**, Hanzhe Wang** [1,2]**, Hongliang Wang** [1,2,*]**, An Luo** [1,2] **and Minying Li** [1]

1   Guangdong Zhicheng Champion Group Co., Ltd., Dongguan 523718, China; weixinwei@hnu.edu.cn (X.W.); zhuxn@hnu.edu.cn (X.Z.); wenyuanzhang@hnu.edu.cn (W.Z.); hanzhewang@hnu.edu.cn (H.W.); an_luo@hnu.edu.cn (A.L.); lmy@zhicheng-champion.com (M.L.)
2   The College of Electrical and Information Engineering, Hunan University, Changsha 410082, China
*   Correspondence: wanghl123@hnu.edu.cn

**Abstract:** The application of the finite control set model predictive control to cascaded inverters is severely limited by its computational complexity. In this paper, a load observer based multilayer model predictive control is proposed for the voltage mode digital power amplifier employing cascaded full-bridge neutral point clamped inverter, which can avoid the use of load current sensor and greatly reduce the controller computation without affecting its dynamic performance. The discrete mathematical model of the voltage mode digital power amplifier employing cascaded full-bridge neutral point clamped inverter is established with filter inductor current and filter capacitor voltage as state variables. A load current observer is designed based on this to avoid the use of load current observer. Based on the discrete model and the observed load current, the upper layer of the multilayer model predictive control determines the optimal level that minimizes the cost function. The middle layer allocates the optimal level to each submodule in order to achieve capacitor voltage balancing. The lower layer determines the switching state of each submodule in order to reduce switching actions. Finally, the experimental results based on the designed nine-level prototype show that the develop multilayer model predictive control lead to acceptable steady state, dynamic and robust performance, with only 1.37% of the run time of the traditional model predictive control.

**Keywords:** model predictive control; cascaded full-bridge neutral point clamped inverter; load observer; calculation amount

## 1. Introduction

Power amplifiers are widely used in industrial testing and measurement [1]. They are also often used to drive underwater acoustic transducers to produce low-frequency tunable sound sources, which can realize submarine navigation and ranging [2]. Currently, power amplifiers can be divided into class A, class B, class AB and class D. Class D power amplifier, also known as the digital power amplifier, is widely used in high-power occasions because of its advantages of low power loss and high efficiency [3]. According to the different types of loads driven by digital power amplifiers, the digital power amplifiers can be divided into two modes: the voltage mode and the current mode. The voltage mode digital power amplifiers are mainly used to provide alternating electrical energy to loads such as piezoelectric ceramic transducers [4]. Compared with two-level or three-level inverter, cascaded inverter, such as cascaded H-bridge inverter [5], modular multilevel converter [6] and cascaded full-bridge neutral point clamped inverter (CFNPCI) [7], has the advantages of high power, low step voltage and step current, strong fault handling ability, modular structure and high output waveform quality; thus, it is often employed by high power digital power amplifiers. However, high power digital power amplifiers employing cascaded inverter use a large number of switching devices, which makes the control the digital power amplifiers very complex. In addition, digital power amplifiers

may be required to output non-sine wave, which will lead to the control methods based on *dq* rotating coordinate system commonly used in ac transmission, and ac motor drive systems arenot suitable for digital power amplifiers.

Aiming at the output closed-loop control problem of the voltage mode digital power amplifier employing cascaded inverter, a single closed-loop PI control is used in [1] for cascaded H-bridge voltage mode digital power amplifier. However, the output gain of PI controller decreases with the increase of frequency, and the parameter adjustment and matching are complex. In [8], a double closed-loop PI control is introduced to control the output voltage of cascaded H-bridge voltage mode digital power amplifier, the control bandwidth is improved compared with the single closed-loop PI controller. However, the parameter adjustment and matching for the double closed-loop PI controller is more complex. Due to the limited dynamic performance of the traditional linear controllers, the sliding mode controller is proposed in [2,9] to control the output voltage of the digital power amplifier, and the dynamic performance is greatly improved. However, it is very difficult to determine the sliding surface of the sliding mode controller, and its engineering application is also limited by the chattering phenomenon [10].

Another kind of nonlinear control, model predictive control (MPC), has been of great interest and widely studied in recent years because of its good dynamic response performance and the ability to handle multiple control objectives at the same time. Moreover, the finite control set MPC has also been successfully applied to cascaded inverters, such as cascaded H-bridge inverter [11–13], modular multilevel converter [14–24] and CFNPCI [25]. The finite control set MPC needs to calculate and compare the cost function for all switching states. However, the number of switching states of cascaded inverters will increase exponentially with the increase of the number of submodules, which leads to a huge amount of computation and severely limits the engineering application of this method. In [14], an MPC of modular multilevel converter with an independent arm-balancing control is proposed to decouple the circulating current and arm voltage with an arm-balancing control. In [15], a novel MPC algorithm with reduced switching frequency is proposed for modular multilevel converters. However, the computation of the controller required by the method in [14,15] is not reduced. Ref. [17] and Ref. [18] respectively reduce the number of effective switching states by removing redundant switching states and grouping, and the computation of the MPC algorithm can be partially reduced. In [19], the allowable fluctuation value of capacitor voltages of modular multilevel converters is considered to reduce the calculation amount of the MPC, but a considerable computation is still unavoidable. In [20–23], hierarchical or multistep control is used to reduce the amount of computation. In [11], a new MPC which selects two voltage levels in a single control period is proposed to reduce the computation burden for cascaded H-bridge inverter. In [13], only the voltage vectors adjacent to the voltage vector used in the previous control period are considered. In [24], only the output levels adjacent to the output level used in the previous control period are considered. Both the method in [13,24] could reduce the number of control options to three, so that the MPC algorithm only needs to cycle three times in each control period, which significantly reduces the calculation of the controller. However, the methods in [11,13,20–24] will affect the dynamic performance of MPC. In addition, the above methods are simplified by reducing the number of control options, and they still cannot avoid the cyclic calculation and comparison of the cost function. In [12], the optimization problem of the MPC is solved by the sphere decoding algorithm. However, the sphere decoding algorithm occupies lots of computing resources all the same.

In this paper, a multilayer MPC (MMPC) is proposed for the voltage mode digital power amplifier employing CFNPCI, which can completely avoid the cyclic calculation and comparison of the cost function without affecting the system dynamic performance, thus greatly reducing the amount of calculation of the controller.

The rest of this paper is organized as follows. In Section 2, the circuit topology of the voltage mode digital power amplifier employing CFNPCI is introduced, and the discrete mathematical model of the digital power amplifier is derived. In order to avoid the use of

load current sensor, a load observer is designed in Section 3, which can estimate the load current online. On this basis, a low complexity MMPC is proposed in Section 4, which divides multiple control objectives of the digital power amplifier system into upper, middle and lower layers. In Section 5, the feasibility and effectiveness of the proposed MMPC are verified on a nine-level experimental prototype.

## 2. Circuit Structure and Discrete Model of the Voltage Mode Digital Power Amplifier Employing CFNPCI

### 2.1. Circuit Structure of the Voltage Mode Digital Power Amplifier Employing CFNPCI

The circuit structure of the voltage mode digital power amplifier employing CFNPCI is shown in Figure 1. The output voltage of the CFNPCI is expressed by $V_{ab}$, and the output voltage of the digital power amplifier, $V_o$, can be obtained after filtering by LC filter composed of filter inductor $L_f$ and filter capacitor $C_f$. The current of the filter inductor is expressed by $i_f$, and its positive direction is shown in Figure 1. The voltage of the filter capacitor is also the digital power amplifier output voltage, which is denoted by $V_o$. The output current of the digital power amplifier is denoted by $i_o$. The CFNPCI consists of $n$ identical submodules in series, and the topology of each submodule is a full-bridge neutral point clamped inverter. The DC side of each submodule uses an isolated power supply module with output voltage of $V_{dc}$ to provide voltage support. Two capacitors with same value, $C_{i1}$ and $C_{i2}$, are connected in series to obtain two electric potentials of $V_{dc}/2$ and $-V_{dc}/2$. Each submodule consists of eightIGBTs, eightanti-parallel diodes and fourclamping diodes. The AC side of each submodule can output five levels of $-2, -1, 0, 1, 2$. Then the voltage mode digital power amplifier composed of $n$ submodules can output $4n + 1$ levels of $-2n, -2n + 1, \ldots, 0, \ldots, 2n - 1$, and 2.

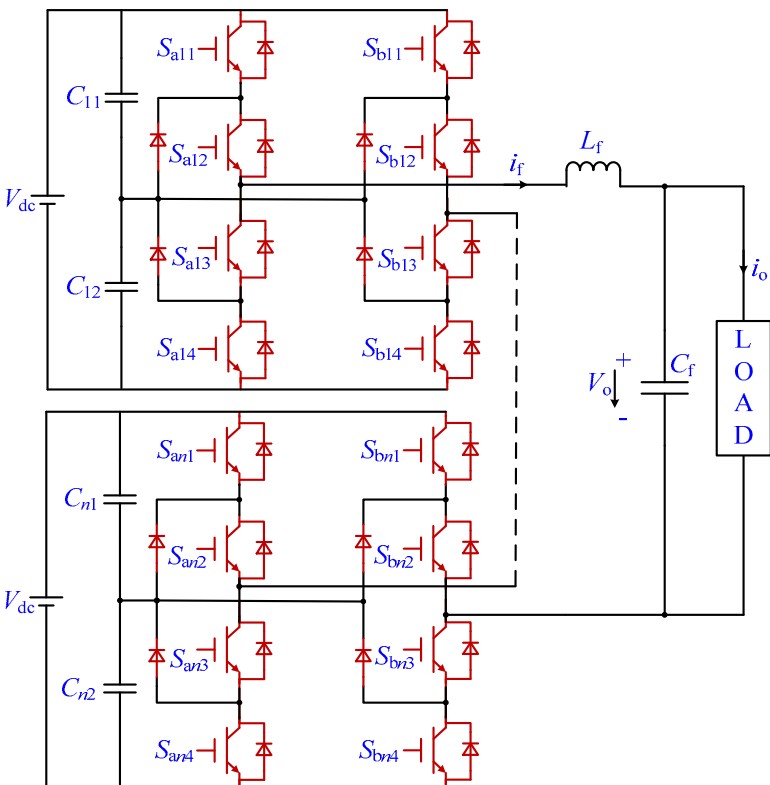

**Figure 1.** The structure of the voltage mode digital power amplifier employing CFNPCI.

The driving signal of IGBT is represented by $S_{xij}$. $x \in \{a, b\}$ denotes legs of the inverter, where $a$ denotes the left one, $b$ denotes the right one. $i \in \{1, 2, \ldots, n\}$ denotes the submodule number, $j \in \{1, 2, 3, 4\}$ denotes the number of the transistor in the same bridge. In normal operation, $S_{ia1}$ and $S_{ia3}$ complement each other, $S_{ia2}$ and $S_{ia4}$ complement each other, too. $S_{ib1}$, $S_{ib2}$, $S_{ib3}$ and $S_{ib4}$ also meet this constraint. $S_i = [S_{ai1}S_{ai2}S_{ai3}S_{ai4}\ S_{bi1}S_{bi2}S_{bi3}S_{bi4}]$ is defined as the switching state of the $i$th submodule, and $M_i$ is used to denote the output level of the $i$th submodule. $U_{Ci1}$ and $U_{Ci2}$ respectively denote the voltages of DC capacitors $C_{i1}$ and $C_{i2}$ in the $i$th submodule. $\Delta U_{Ci}$ is defined as the difference between $U_{Ci1}$ and $U_{Ci2}$, which can be calculated by (1).

$$\Delta U_{Ci} = U_{Ci1} - U_{Ci2} = 2U_{Ci1} - V_{dc} \tag{1}$$

Table 1 shows the relationship between the change of $\Delta U_{Ci}$, the output level $M_i$, the inductor current $i_f$, and the nineeffective switching states.

**Table 1.** Relationship between $\Delta U_{Ci}$, $M_i$, $i_f$ and $S_i$.

| $M_i$ | $S_I$ | $\Delta U_{Ci}$ | |
|---|---|---|---|
| | | $i_f > 0$ | $i_f < 0$ |
| 2 | S1 = [1 1 0 0 0 0 1 1] | invariant | invariant |
| 1 | S2 = [1 1 0 0 0 1 1 0] | decrease | increase |
| | S3 = [0 1 1 0 0 0 1 1] | increase | decrease |
| 0 | S4 = [1 1 0 0 1 1 0 0] | invariant | invariant |
| | S5 = [0 1 1 0 0 1 1 0] | invariant | invariant |
| | S6 = [0 0 1 1 0 0 1 1] | invariant | invariant |
| −1 | S7 = [0 1 1 0 1 1 0 0] | increase | decrease |
| | S8 = [0 0 1 1 0 1 1 0] | decrease | increase |
| −2 | S9 = [0 0 1 1 1 1 0 0] | invariant | invariant |

## 2.2. Discrete Model of the Voltage Mode Digital Power Amplifier Employing CFNPCI

Assuming that the capacitor voltages in all submodules are well balanced, (2) can be obtained according to Kirchhoff's law of voltage and current.

$$\begin{cases} L_f \frac{di_f}{dt} + V_o = V_{ab} \\ C_f \frac{dV_o}{dt} + i_o = i_f \end{cases} \tag{2}$$

In (2), $V_{ab}$ can be calculated by (3).

$$V_{ab} = \frac{V_{dc}}{2} \sum_{i=1}^{n} M_i \tag{3}$$

The total output level $M \in \{-2n, -2n+1, \ldots, 0, \ldots, 2n-1, 2\}$, and can be expressed as (4)

$$M = \sum_{i=1}^{n} M_i \tag{4}$$

Using $x = [i_f\ V_o]^T$ to denote the system state variables, and substituting (3) and (4) into (2), the continuous system model of the voltage mode digital power amplifier can be obtained, as shown in (5),

$$\dot{x} = Ax + B_1 M + B_2 i_o \tag{5}$$

where $A = \begin{bmatrix} 0 & -\frac{1}{L_f} \\ \frac{1}{C_f} & 0 \end{bmatrix}$, $B_1 = \begin{bmatrix} \frac{V_{dc}}{2L_f} \\ 0 \end{bmatrix}$, $B_2 = \begin{bmatrix} 0 \\ -\frac{1}{C_f} \end{bmatrix}$.

For the purpose of digital control, (5) should be discretized. The sampling period is denoted as $T_S$. The discrete model is expressed as (6),

$$x(k+1) = A_{\mathrm{d}}x(k) + B_{1\mathrm{d}}M(k) + B_{2\mathrm{d}}i_{\mathrm{o}}(k) \tag{6}$$

where $A_{\mathrm{d}} = e^{AT_S}$, $B_{1\mathrm{d}} = \int_0^{T_S} e^{A\tau}B_1 d\tau$, $B_{2\mathrm{d}} = \int_0^{T_S} e^{A\tau}B_2 d\tau$, $k$ and $k+1$ represent the $kT_S$ and $(k+1)T_S$ instant, respectively.

In addition, the system output equation can be expressed by (7).

$$V_{\mathrm{o}}(k) = \begin{bmatrix} 0 & 1 \end{bmatrix} x(k) \tag{7}$$

## 3. Design of the Load Observer

The load current $i_{\mathrm{o}}$, which is also the output current of the voltage mode digital power amplifier, depends on the load. It can be seen from (6) that $i_{\mathrm{o}}$ is an interference term for the close-loop of $V_{\mathrm{o}}$, which is generally measured by the configured current sensor. In this paper, a state observer is designed to estimate $i_{\mathrm{o}}$ online, which avoids the use of current sensor and reduces the system hardware cost.

Assuming that the load current $i_{\mathrm{o}}$ is constant in a single sampling period [10], and extending $i_{\mathrm{o}}$ to a state variable of the system, (8) can be obtained.

$$\begin{bmatrix} x(k) \\ i_{\mathrm{o}}(k) \end{bmatrix} = \begin{bmatrix} A_{\mathrm{d}} & B_{2\mathrm{d}} \\ 0 & 1 \end{bmatrix} \begin{bmatrix} x(k-1) \\ i_{\mathrm{o}}(k-1) \end{bmatrix} + \begin{bmatrix} B_{1\mathrm{d}} \\ 0 \end{bmatrix} M(k-1) \tag{8}$$

The system output equation can be rewritten as (9).

$$Y(k) = \begin{bmatrix} i_{\mathrm{f}}(k) \\ V_{\mathrm{o}}(k) \end{bmatrix} = \begin{bmatrix} 1 & 0 & 0 \\ 0 & 1 & 0 \end{bmatrix} \begin{bmatrix} i_{\mathrm{f}}(k) \\ V_{\mathrm{o}}(k) \\ i_{\mathrm{o}}(k) \end{bmatrix} \tag{9}$$

Let $\Phi = \begin{bmatrix} A_{\mathrm{d}} & B_{2\mathrm{d}} \\ 0 & 1 \end{bmatrix}$, $G = \begin{bmatrix} B_{1\mathrm{d}} \\ 0 \end{bmatrix}$, $C = \begin{bmatrix} 1 & 0 & 0 \\ 0 & 1 & 0 \end{bmatrix}$, then the state observer shown in (10) can be constructed to realize online estimation of load current $i_{\mathrm{o}}$.

$$\hat{X}(k) = \Phi\hat{X}(k-1) + GM(k-1) + K\left[Y(k) - \hat{Y}(k)\right] \tag{10}$$

In (10), $\hat{X}(k) = \begin{bmatrix} \hat{i}_{\mathrm{f}}(k) & \hat{V}_{\mathrm{o}}(k) & \hat{i}_{\mathrm{o}}(k) \end{bmatrix}$ denotes the estimated value of the extended system state variable $X(k) = \begin{bmatrix} i_{\mathrm{f}}(k) & V_{\mathrm{o}}(k) & i_{\mathrm{o}}(k) \end{bmatrix}$, and $\hat{Y}(k)$ is the estimated value of $Y(k)$. $K$ is the observer gain matrix.

$\hat{i}_{\mathrm{o}}(k)$ is the estimated value of $i_{\mathrm{o}}(k)$, and can be calculated based on (11).

$$\hat{i}_{\mathrm{o}}(k) = \begin{bmatrix} 0 & 0 & 1 \end{bmatrix} \hat{X}(k) \tag{11}$$

The designed load observer can be regarded as a discrete Kalman Filter, and its gain matrix $K$ can be calculated to obtain the optimal estimation of $i_{\mathrm{o}}$ in the presence of random measurement noise.

It should be pointed out that the estimation error of the designed load observer depends on the selected gain matrix $K$. Unreasonable gain parameters will lead to large estimation error of $i_{\mathrm{o}}$, and eventually lead to tracking error of the MMPC in Section 4. Therefore, the gain matrix of the load observer should be elaborated to achieve a tradeoff between the dynamic performance and the noise immunity. The parameter selection method in [10] can be used to determine the gain matrix $K$.

## 4. Multilayer Model Predictive Control for the Voltage Mode Digital Power Amplifier Employing CFNPCI

In traditional model predictive control (TMPC), the cost function under all switching states must be calculated in each control cycle, and the switching state which minimizes the cost function should be selected. However, if we extend the TMPC to CFNPCI with $n$ submodules, there will be $9^n$ effective switching states. This also means that the cost function needs to be repeatedly calculated for $9^n$ times to obtain the optimal switching state, and the amount of calculation increases exponentially with the increase of the submodule number $n$, which will pose a great challenge to the computational performance of the controller.

The overall control of the CFNPCI, as shown in Figure 2a, consists of the designed load observer and the proposed MMPC. The structure of the proposed MMPC is shown in Figure 2, including upper, middle and lower layers. No matter how large $n$ is, the MMPC can avoid the repeated calculation of the cost function, greatly reduce the computational burden of the controller, and the dynamic performance can also not be affected. The upper layer control is used to determine the optimal level to minimize the cost function. The middle layer control is used to distribute the optimal level to each submodule aiming to balance the capacitor voltages in all the submodules. The lower layer control is used to determine the switching state of each submodule aiming to minimize the switching actions.

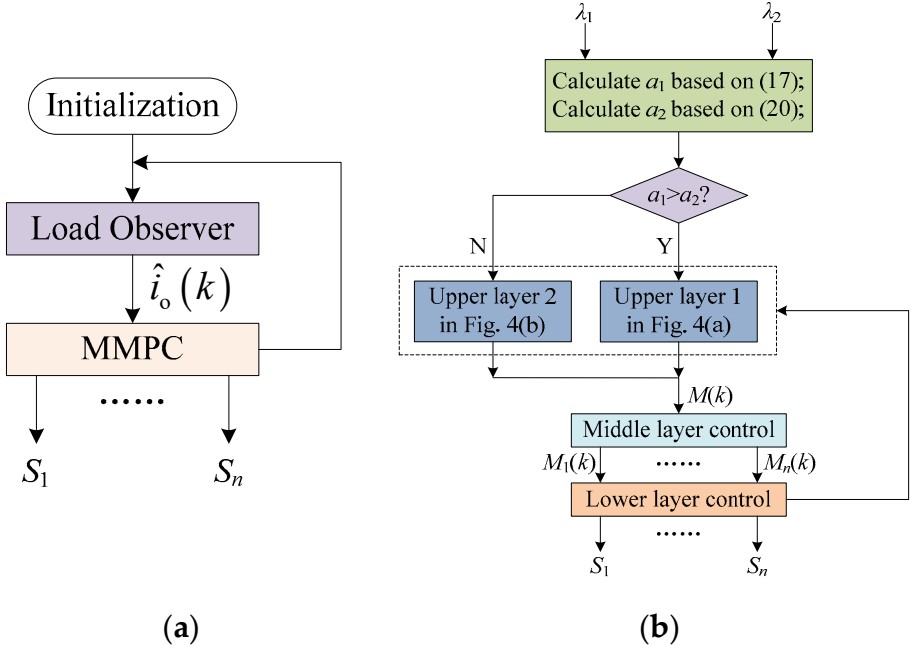

**(a)**

**(b)**

**Figure 2.** Structure diagram of the load observer based MMPC. (**a**) Structure of the overall control. (**b**) Structure of the proposed MMPC.

### 4.1. The Upper Layer Control

$V_{\text{oref}}(k + 1)$ is used to represent the output voltage reference at $(k + 1)T_{\text{S}}$. Then, the filter inductor current reference $i_{\text{fref}}(k + 1)$ can be calculated by (12).

$$i_{\text{fref}}(k + 1) = C_{\text{f}} \frac{V_{\text{oref}}(k + 1) - V_{\text{oref}}(k)}{T_{\text{S}}} + \hat{i}_{\text{o}}(k) \tag{12}$$

The used system cost function $J(h)$ can be expressed by (13),

$$J(h) = \lambda_1 |i_{\text{fref}}(k + 1) - i_{\text{fh}}(k + 1)| + \lambda_2 |V_{\text{oref}}(k + 1) - V_{\text{oh}}(k + 1)| \\ h \in H = \{-2n, -2n + 1, \cdots, 0, \cdots, 2n - 1, 2n\} \tag{13}$$

where $i_{\mathrm{fh}}(k+1)$ and $V_{\mathrm{oh}}(k+1)$ respectively denote the instantaneous values of the inductor current $i_{\mathrm{f}}$ and the capacitor voltage $V_{\mathrm{oh}}$ at $(k+1)T_{\mathrm{s}}$ if the candidate output level $h$ is selected in the $k$th control period. $\lambda_1$ and $\lambda_2$ are the weight factors.

$i_{\mathrm{fh}}(k+1)$ can be calculated based on (14), while $V_{\mathrm{oh}}(k+1)$ can be calculated based on (15),

$$i_{\mathrm{fh}}(k+1) = A_{\mathrm{d11}}i_{\mathrm{f}}(k) + A_{\mathrm{d12}}V_{\mathrm{o}}(k) + B_{\mathrm{1d11}}h + B_{\mathrm{2d11}}\hat{i}_{\mathrm{o}}(k) \tag{14}$$

$$V_{\mathrm{oh}}(k+1) = A_{\mathrm{d21}}i_{\mathrm{f}}(k) + A_{\mathrm{d22}}V_{\mathrm{o}}(k) + B_{\mathrm{1d21}}h + B_{\mathrm{2d21}}\hat{i}_{\mathrm{o}}(k) \tag{15}$$

where $\hat{i}_{\mathrm{o}}(k)$ is obtained from the designed load observer in Section 3.

Define the function $J_1(h)$ as shown in (16),

$$\begin{aligned} J_1(h) &= \lambda_1|i_{\mathrm{fref}}(k+1) - i_{\mathrm{fh}}(k+1)| \\ &= \left|\lambda_1 B_{\mathrm{1d11}}h + \lambda_1 A_{\mathrm{d11}}i_{\mathrm{f}}(k) + \lambda_1 A_{\mathrm{d12}}V_{\mathrm{o}}(k) + \lambda_1 B_{\mathrm{2d11}}\hat{i}_{\mathrm{o}}(k) - \lambda_1 i_{\mathrm{fref}}(k+1)\right| \\ &= a_1|h - h_1| \end{aligned} \tag{16}$$

where $a_1$ can be calculated by (17), $h_1$ can be calculated by (18).

$$a_1 = |\lambda_1 B_{\mathrm{1d11}}| \tag{17}$$

$$h_1 = \frac{i_{\mathrm{fref}}(k+1) - A_{\mathrm{d11}}i_{\mathrm{f}}(k) - A_{\mathrm{d12}}V_{\mathrm{o}}(k) - B_{\mathrm{2d11}}\hat{i}_{\mathrm{o}}(k)}{B_{\mathrm{1d11}}} \tag{18}$$

Define the function $J_2(h)$ as shown in (19),

$$\begin{aligned} J_2(h) &= \lambda_2|V_{\mathrm{oref}}(k+1) - V_{\mathrm{oh}}(k+1)| \\ &= \left|\lambda_2 B_{\mathrm{1d21}}h + \lambda_2 A_{\mathrm{d21}}i_{\mathrm{f}}(k) + \lambda_2 A_{\mathrm{d22}}V_{\mathrm{o}}(k) + \lambda_2 B_{\mathrm{2d21}}\hat{i}_{\mathrm{o}}(k) - \lambda_2 V_{\mathrm{oref}}(k+1)\right| \\ &= a_2|h - h_2| \end{aligned} \tag{19}$$

where $a_2$ can be calculated by (20), $h_2$ can be calculated by (21).

$$a_2 = |\lambda_2 B_{\mathrm{1d21}}| \tag{20}$$

$$h_2 = \frac{V_{\mathrm{oref}}(k+1) - A_{\mathrm{d21}}i_{\mathrm{f}}(k) - A_{\mathrm{d22}}V_{\mathrm{o}}(k) - B_{\mathrm{2d21}}\hat{i}_{\mathrm{o}}(k)}{B_{\mathrm{1d21}}} \tag{21}$$

It is easy to know that $J(h) = J_1(h) + J_2(h)$. If we assume that $h$ is a continuous variable whose domain of definition is $(-\infty, +\infty)$, according to (16) and (19), the relationship between $J_1(h)$ and $h$ is linear, the relationship between $J_2(h)$ and $h$ also is linear.

Therefore, the function curves of $J(h)$, $J_1(h)$ and $J_2(h)$ with respect to $h$ have only four cases, as shown in Figure 3.

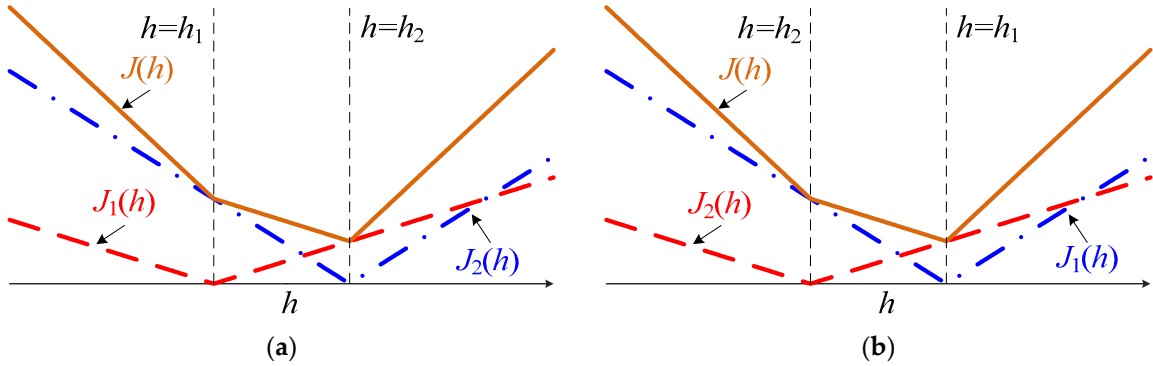

**Figure 3.** *Cont.*

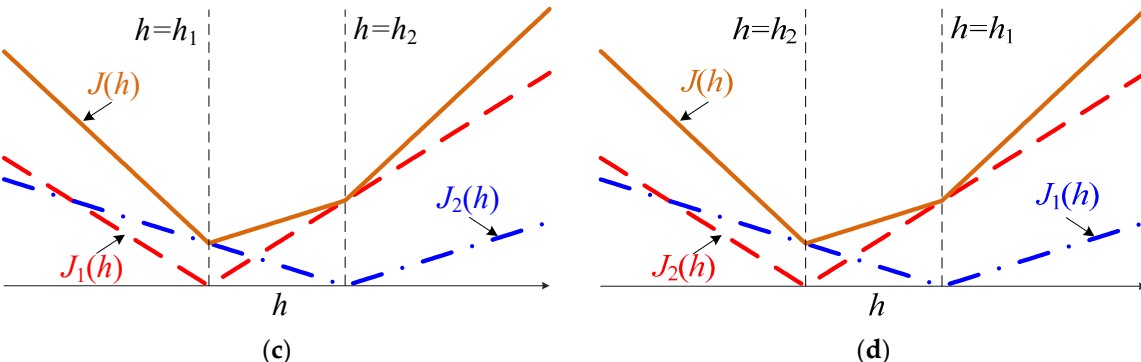

**Figure 3.** Functional curves of $J(h)$, $J_1(h)$ and $J_2(h)$: (**a**) $a_2 > a_1$, $h_1 < h_2$. (**b**) $a_2 < a_1$, $h_1 > h_2$. (**c**) $a_2 < a_1$, $h_1 < h_2$. (**d**) $a_2 > a_1$, $h_1 > h_2$.

Based on Figure 3, we can discuss the abscissa $p$ when $J(h)$ takes the minimum value in the interval $[-2n, 2n]$. The results are shown in Table 2.

The optimal output level $M(k)$ must be the element in the set $H$ that minimizes the cost function $J(h)$, while the minimum of $J(h)$ should be 0. Therefore, $M(k)$ can be calculated by (22),

$$M(k) = \begin{cases} floor(p), & \text{if } J[floor(p)] < J[ceil(p)] \\ ceil(p), & \text{if } J[floor(p)] \geq J[ceil(p)] \end{cases} \tag{22}$$

where $floor(p)$ means to round down $p$ to obtain the largest integer not greater than $p$; $ceil(p)$ means to round up $p$ to obtain the smallest integer not less than $p$.

**Table 2.** The abscissa of the minimum of $J(h)$ in $[-2n, 2n]$.

| $h_1$ and $h_2$ ＼ $a_1$ and $a_2$ | $a_2 > a_1$ | $a_2 \leq a_1$ |
|---|---|---|
| $h_1 < -2n$, $h_2 < -2n$ | $p = -2n$ | $p = -2n$ |
| $-2n \leq h_1 \leq 2n$, $h_2 < -2n$ | $p = -2n$ | $p = h_1$ |
| $h_1 > 2n$, $h_2 < -2n$ | $p = -2n$ | $p = 2n$ |
| $h_1 < -2n$, $-2n \leq h_2 \leq 2n$ | $p = h_2$ | $p = -2n$ |
| $-2n \leq h_1 \leq 2n$, $-2n \leq h_2 \leq 2n$ | $p = h_2$ | $p = h_1$ |
| $h_1 > 2n$, $-2n \leq h_2 \leq 2n$ | $p = h_2$ | $p = 2n$ |
| $h_1 < -2n$, $h_2 > 2n$ | $p = 2n$ | $p = -2n$ |
| $-2n \leq h_1 \leq 2n$, $h_2 > 2n$ | $p = 2n$ | $p = h_1$ |
| $h_1 > 2n$, $h_2 > 2n$ | $p = 2n$ | $p = 2n$ |

In order to further reduce the amount of calculation, the approximate calculation method shown in (23) is used in this paper,

$$M(k) \approx round(p) \tag{23}$$

where $round(p)$ means to round $p$ to obtain the nearest integer to $p$.

The maximum difference between the result of (23) and the result of (22) is one level, which will not cause unacceptable control error. However, if we obtain $M(k)$ based on (23) rather than (22), $J[floor(p)]$ and $J[ceil(p)]$ can be avoided calculating and comparing with each other.

Because $a_1$ and $a_2$ are only determined by the weight factors and system parameters, they can be calculated off-line in the initialization link, and there is no need to calculate them in each control cycle, so the algorithm computation can be further reduced. Figure 4 shows the flow chart of the upper layer control when $a_1 \geq a_2$ and $a_1 < a_2$ respectively.

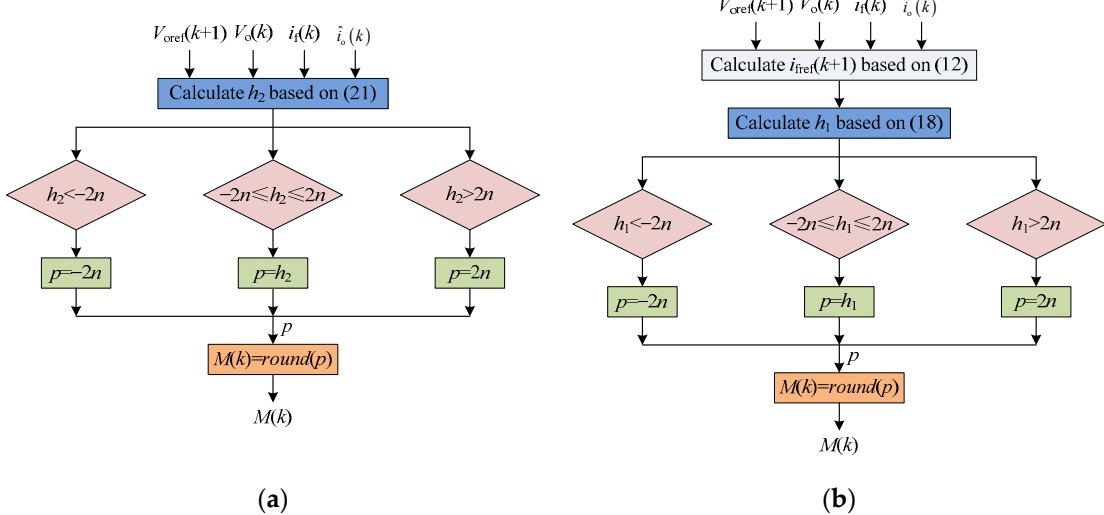

(**a**)  (**b**)

**Figure 4.** Flow chart of the upper control algorithm: (**a**) $a_2 \geq a_1$. (**b**) $a_2 < a_1$.

Based on the above analysis, the tracking error of the MMPC is not only related to the estimation accuracy of the output current $i_o$, but also related to its control period. A shorter control period will lead to a smaller tracking error. The upper layer control significantly reduces the computation and the running time of the MPC algorithm, so that a shorter control period can be used and a smaller tracking error can be obtained.

*4.2. The Middle Layer Control*

Since each submodule can output fivelevels of 2, 1, 0, −1, −2, the constraint condition of (24) must be satisfied when the optimal level obtained by upper layer control is allocated to each submodule.

$$\begin{cases} \sum_{i=1}^{n} M_i(k) = M(k) \\ M_i(k) \in \{-2, -1, 0, 1, 2\} \end{cases}, i = 1, 2, \cdots, n \tag{24}$$

However, the number of level combinations satisfying the constraint of (24) is large, which makes the level allocation complex. In this paper, the following three constraints are added to simplify the level allocation algorithm:

(1) When $M(k) > 0$, $M_i(k)$ is selected from 2, 1 and 0;

(2) When $M(k) < 0$, $M_i(k)$ is selected from −2, −1 and 0;

(3) When $M(k) = 0$, $M_i(k)$ is set to 0.

Furthermore, the analysis of Table 1 shows that only the 1 and −1 levels will affect the submodule capacitor voltages. Therefore, in order to achieve capacitor voltage balance, the submodules with larger capacitor voltage differences should be allocated 1 or −1 level, while the submodules with smaller capacitor voltage differences should be allocated 2, 0 or −2 level. Based on this analysis, this paper proposes the following allocation steps:

Step 1: Calculate the absolute values of capacitor voltage differences of all submodules;

Step 2: Sort the absolute values of capacitor voltage differences of all submodules in descending order;

Step 3: According to the sorting result of Step 2, the first allocation pass is performed from front to back, and each submodule is allocated +1 level ($M_i(k) > 0$) or −1 level ($M_i(k) < 0$). If the output level cannot be allocated completely in the first pass, the second pass will be performed from the back to the front until the output level is allocated completely.

Figure 5 shows the level allocation process when the submodule number $n = 3$ and the total output level $M(k) = 2$ or −5. It can be seen that by using the proposed level allocation steps, the submodules with larger capacitor voltage differences can be more likely to be

allocated 1 or −1 level, and the submodules with smaller capacitor voltage differences can be more likely to be allocated 2, 0 or −2 level, thus creating conditions for capacitor voltage balance control.

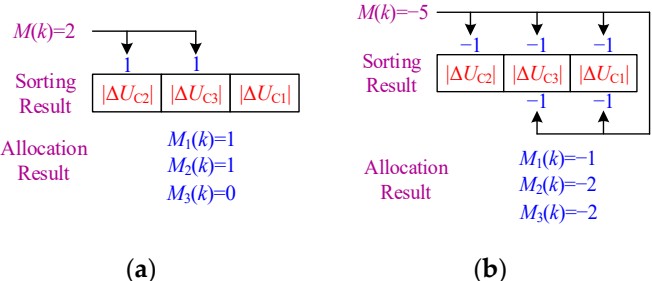

**Figure 5.** Output level allocation schematic: (**a**) $M(k) = 2$. (**b**) $M(k) = -5$.

### 4.3. The Lower Layer Control

Due to the additional level allocation constraints in the middle layer control, when $M(k) > 0$, the output level number $M_i(k)$ of the $i$th submodule can only switch between 0, 1 and 2.

Table 3 counts the IGBT action times when the three levels switch with each other. At the same time, according to Table 1, if a submodule is allocated +2 level, the corresponding switching state can only select S1. If a submodule is allocated +1 level, in order to achieve capacitor voltage balance, S2 should be selected when the signs of $i_o$ and $\Delta U_{Ci}$ are the same, while S3 should be selected when they are opposite. If a submodule is allocated 0 level, in order to minimize the number of switching actions when switching between 0 level and +1 level, S5 should be selected.

**Table 3.** The number of switching action when the output level switched between 0, 1 and 2.

| Switching Level | Switching State | Switching Actions |
|:---:|:---:|:---:|
| 2 and 1 | S1 and S2 | 2 |
|  | S1 and S3 | 2 |
| 1 and 0 | S2 and S4 | 2 |
|  | S2 and S5 | 2 |
|  | S2 and S6 | 6 |
|  | S3 and S4 | 6 |
|  | S3 and S5 | 2 |
|  | S3 and S6 | 2 |
| 2 and 0 | S1 and S4 | 4 |
|  | S1 and S5 | 4 |
|  | S1 and S6 | 4 |

When $M_i(k) < 0$, the following conclusions can be drawn: (1) if a submodule is allocated −2 level, S9 should be selected; (2) if a submodule is allocated −1 level, S8 should be selected when the signs of $i_o$ and $\Delta U_{Ci}$ are the same, while S7 should be selected when they are opposite; (3) if a submodule is allocated 0 level, S5 should be selected.

Figure 6 shows the flow chart of the switching state selection process. It can be seen that the middle layer control and the lower layer control are parallel structure, so they are more suitable to be implemented in FPGA.

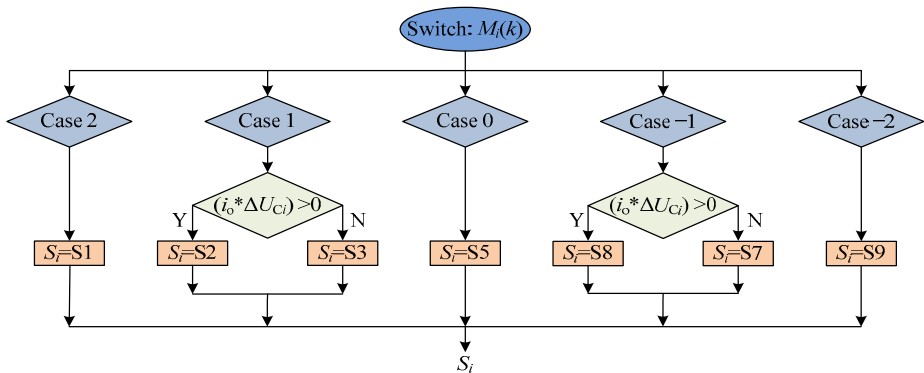

**Figure 6.** Flow chart of the switching state selection process.

### 4.4. Comparisons of Computation

In order to prove the advantages of the MMPC in reducing the calculation amount, the TMPC proposed in [7] is used to compare with the algorithm in this paper. In order to obtain a general conclusion independent of the specific controller, Table 4 compares the number of addition, subtraction, multiplication, division and comparison operations required by the TMPC and the MMPC.

It can be seen from Table 4 that there is a linear or quadratic relationship between the calculation amount required by the MMPC and the submodule number $n$, while the calculation amount required by the TMPC has an exponential relationship with $n$. Therefore, the MMPC proposed in this paper has a significant advantage in the algorithm computation.

**Table 4.** Comparisons of computation complexity.

| Control Algorithm | | Addition | Subtraction | Multiplication | Division | Comparison |
|---|---|---|---|---|---|---|
| **TMPC** | | $(8n + 7) \times 9^n$ | $(7n + 2) \times 9^n$ | $(12n + 10) \times 9^n$ | $(4n + 1) \times 9^n$ | $(2n + 1) \times 9^n$ |
| MMPC | Upper Layer | 1 | 4 | 4 | 1 | 2 |
| | Middle Layer | $\lvert M \rvert$ | $N + \lvert M \rvert$ | 0 | 0 | $0.5n^2 + 0.5n + 1$ |
| | Lower Layer | 0 | 0 | $n$ | 0 | $2n$ |
| | Total | $\lvert M \rvert + 1$ | $N + 3 + \lvert M \rvert$ | $N + 3$ | 0 | $0.5n^2 + 2.5n + 4$ |

## 5. Experimental Verification

In order to verify the feasibility and effectiveness of the designed load current observer and the proposed MMPC in this paper, an experimental prototype as shown in Figure 7 is built in the laboratory for experimental verification. The experimental prototype consists of two submodules, which can output up to ninelevels. The output frequency band of the prototype is from 50 Hz to 800 Hz. The filter inductance is 2 mH, and the filter capacitance is 4.7 μF. The rated power of each submodule is 2 kW, the dc input voltage f each submodule is 300 V and the dc capacitance is 1070 μF. The control period is set to 25 μs.

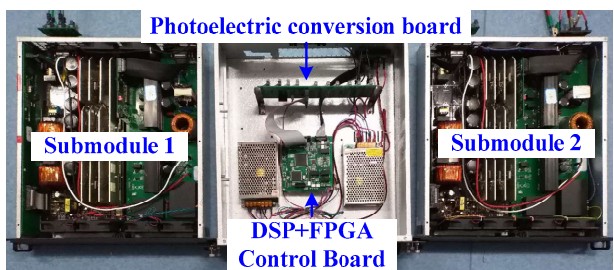

**Figure 7.** Diagram of the experimental prototype.

In the designed experimental prototype, the number of submodules $n$ is equal to 2, and the maximum total output level $M$ can be taken as 4. Based on Table 4, the TMPC requires 1863 addition calculations, 1296 subtraction calculations, 2754 multiplication calculations, 729 division calculations and 405 comparison calculations for each control period, which poses a huge challenge to the computing performance of the controller. However, the MMPC proposed in this paper only requires 5 addition calculations, 9 subtraction calculations, 5 multiplication calculations, and 11 comparison calculations for each control period. In addition, both the TMPC and the MMPC are tested on a DSP of TMS320F28335. The results show that the TMPC takes 503 μs to run once, while the improved algorithm only takes 6.9 μs to run once. Therefore, the develop multilayer structure greatly reduces the computation burden of the controller and can be acceptable for general digital controllers.

### 5.1. Steady State Performance

In order to study the steady state performance of the designed load observer and the proposed MMPC, the output voltage reference $V_{oref}$ is set as a sine wave with frequency of 800 Hz and amplitude of 550 V. The load is set to an 80 Ω resistor. The experiment results are shown in Figure 8, where Figure 8a shows the waveforms of $V_o$ and its reference $V_{oref}$, Figure 8b shows the waveforms of $i_f$ and its reference $i_{fref}$, Figure 8c shows the waveforms of $i_o$ and its estimated value, and Figure 8d shows the capacitor voltages of the first submodule. It should be pointed out that the waveform of $i_o$ in Figure 8c is obtained by directly measuring based on the current probe.

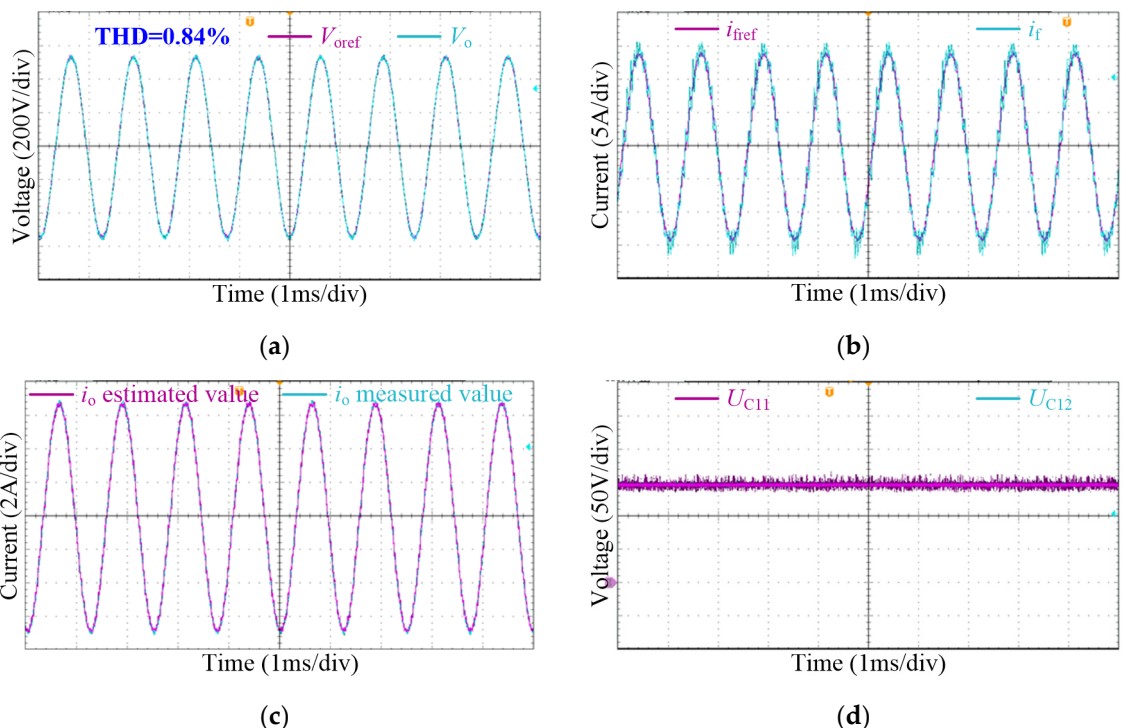

**Figure 8.** Steady state experimental results: (**a**) Waveforms of $V_o$ and $V_{oref}$. (**b**) Waveforms of $i_f$ and $i_{fref}$. (**c**) Waveforms of the measured value and the estimated value of $i_o$. (**d**) Waveforms of capacitor voltages.

It can be seen from Figure 8 that the designed load observer accurately estimates the actual load current $i_o$ under steady state condition. The small estimating error of $i_o$ leads the proposed MMPC to output accuracy inductor current and output voltage, which means small tracking errors are obtained. In addition, the total harmonic distortion rate of the output voltage is tested to be 0.84%, which shows acceptable waveform quality. Therefore, both the load observer and the MMPC show good steady state performance.

### 5.2. Dynamic Performance

The output voltage reference $V_{oref}$ is set as a sine wave with frequency of 50Hz and amplitude of 275 V, while its amplitude steps to 550 V at some point. The load is still set to an 80 $\Omega$ resistor. Under this condition, the dynamic performance of the load observer and the MMPC is tested. The experiment results are shown in Figure 9, where Figure 9a shows the waveforms of $V_o$ and its reference $V_{oref}$, Figure 9b shows the waveforms of $i_f$ and its reference $i_{fref}$, Figure 9c shows the waveforms of $i_o$ and its estimated value, and Figure 9d shows the capacitor voltages of the first submodule.

It can be seen from Figure 9 that, within 0.4 ms, the designed load observer can still quickly and accurately estimate the actual load current under the condition of both the output voltage reference and the load current step change. This also means that the selected gain parameters of the load observer have a good tradeoff between the dynamic performance and the noise immunity. With the help of this, the proposed MMPC also quickly and accurately tracks the step change output reference. Therefore, the dynamic performances of both the load observer and the MMPC are verified.

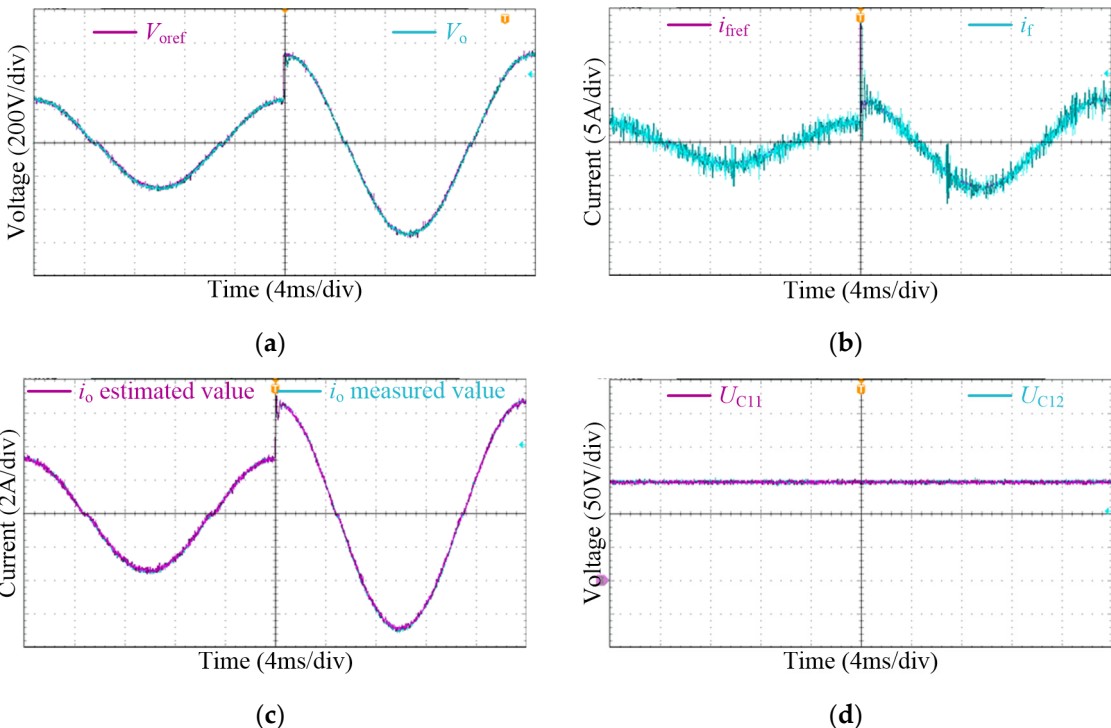

**Figure 9.** Dynamic state experimental results: (**a**) Waveforms of $V_o$ and $V_{oref}$. (**b**) Waveforms of $i_f$ and $i_{fref}$. (**c**) Waveforms of the measured value and the estimated value of $i_o$. (**d**) Waveforms of capacitor voltages.

### 5.3. Robust Performance

The load of the voltage mode digital power amplifier employing CFNPCI is set as the uncontrolled rectifier bridge shown in Figure 10, which is a nonlinear load. In the rectifier bridge, the capacitance is set to be 400 µF, the resistance is set to be 120 $\Omega$. The robust performance of the load observer and the MMPC is verified under this condition. The output voltage reference $V_{oref}$ is still set to a sine wave with frequency of 50 Hz and amplitude of 550 V.

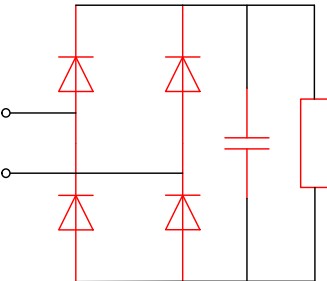

**Figure 10.** Uncontrollable rectifier bridge load.

Figure 11 shows the waveforms of $V_{\mathrm{o}}$ and its reference $V_{\mathrm{oref}}$. It can be seen that, even when the voltage mode digital power amplifier supplies power to a non-linear load, the total harmonic distortion rate of the output voltage still reaches 1.74%, which shows strong system robustness.

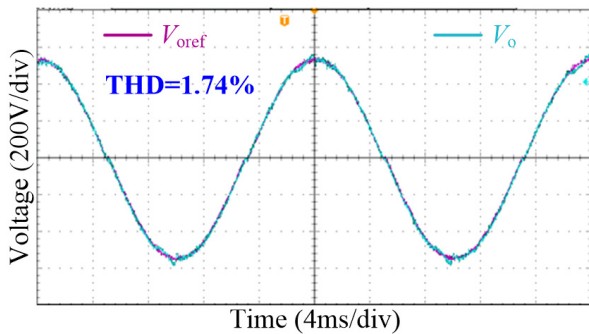

**Figure 11.** Robustness experimental results.

## 6. Conclusions

A cascaded multilevel inverter provides an effective way to increase the capacity of the digital power amplifier. In this paper, a load observer is designed for the voltage mode digital power amplifier employing CFNPCI topology, which could estimate the output current online without configuring the output current sensor, thus reducing the hardware cost. On this basis, an MMPC is proposed, which could completely avoid cyclic calculation and comparison of the system cost function, so the amount of calculation of the controller can be significantly reduced. The test results show that the TMPC takes 503 μs to run once, while the MMPC only takes 6.9 μs to run once. In addition, the experimental results show that the total harmonic distortion rate of the output voltage is only 0.84% in the steady state experiment, both the load observer and the MMPC could track the step change load and reference within 0.4ms in the dynamic experiment, and the total harmonic distortion rate of the output voltage can still reach 1.74% even when the digital power amplifier supplies power for diode rectifier bridge load.

**Author Contributions:** Conceptualization, X.W.; formal analysis, A.L.; funding acquisition, M.L.; investigation, X.W.; methodology, A.L.; project administration, M.L.; software, W.Z.; validation, X.Z. and H.W. (Hanzhe Wang); writing-original draft, X.W.; writing—review and editing, H.W. (Hongliang Wang) and A.L. All authors have read and agreed to the published version of the manuscript.

**Funding:** The Program for Guangdong Introducing Innovative and Entrepreneurial Teams: 2017ZT07G23; Research on High-power and High-efficiency Electro-acoustic Transduction Mechanism and Control Method: 51837005; Research on Topology, Passive Current Sharing Mechanism and Control for Multiphase Resonant Converter with Coupled Resonant Tank: 51977069.

**Conflicts of Interest:** The authors declare no conflict of interest.

## Abbreviations

All the parameters related to equations are defined in the following table.

| | |
|---|---|
| $U_{Ci1}$ | Capacitor voltage of $C_{i1}$ |
| $U_{Ci2}$ | Capacitor voltage of $C_{i2}$ |
| $V_{dc}$ | Voltage of Submodule dc power supply |
| $L_f$ | Filter inductor |
| $C_f$ | Filter capacitor |
| $i_f$ | Current of the $i_f$ |
| $V_o$ | Voltage of $C_f$ |
| $V_{ab}$ | Output voltage of the CFNPCI |
| $M_i$ | Output level of Submodule $i$ |
| $M$ | Total output level |
| $x$ | System state variable |
| $i_o$ | Output current |
| $X$ | Extended state variable |
| $Y$ | Output variable of the load observer |
| $\hat{X}$ | Estimated value of $X$ |
| $\hat{Y}$ | Estimated value of $Y$ |
| $\hat{i}_f$ | Estimated value of $i_f$ |
| $\hat{V}_o$ | Estimated value of $V_o$ |
| $\hat{i}_o$ | Estimated value of $i_o$ |
| $K$ | Gain matrix of the load observer |
| $V_{oref}$ | Control reference of $V_o$ |
| $I_{fref}$ | Control reference of $i_f$ |
| $J(h)$ | Total cost function |
| $J_1(h)$ | Cost function for filter inductor current control |
| $J_2(h)$ | Cost function for output voltage control |
| $h_1$ | Solution of $J_1(h) = 0$ |
| $h_2$ | Solution of $J_2(h) = 0$ |
| $\lambda_1$ | Weight factor for filter inductor current control |
| $\lambda_2$ | Weight factor for output voltage control |
| $T_S$ | Control period |

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
