# Peer review of "Multilayer Model Predictive Control for a Voltage Mode Digital Power Amplifier"

_electronics, doi:10.3390/electronics10141699_

Round 1

Reviewer 1 Report

  1. Interesting research in a very well-prepared manuscript that needs some mild revisions.
  2. Abstract is okay but is not likely to entice the readership to continue reading the rest of the manuscript.
    • Use of acronyms/abbreviations in an abstract is unlikely to attract readers not already aware of the manuscript’s content. Consider whether the reader gains anything by seeing in the abstract FCS-MPC, MMPC, DPA, NPC, and CFNPCI.
    • Results are only presented in weak, qualitative fashion. Highest quality expression of main conclusions or interpretations is quantitative results discussed in the broadest context possible, e.g., percent performance improvement compared to a declared benchmark. “…feasibility and effectiveness are verified…” is very weakly stated results compared to “…xxx percent performance improvement over conventional methods was achieved….”. Seeking quantitative results in the conclusions, the reader only finds “..good steady state performance…”. Meanwhile, line 341 contains quantitative assessment, “…total harmonic distortion of 0.84%....”.
  3. Introduction is decently done with some omitted very recent literature and abuse of multi-citation without elaboration.
    • Deterministic artificial intelligence (as recently applied to discrete and continuous mathematical models of DC motor circuits) was omitted as a competing alternative that is significantly less complex than MPC.
    • Whiplash compensation (as recently applied to space robotics) whose provenance lies in optimization was omitted as a competing alternative that uses Pontryagin’s minimization of Hamiltonian systems to derive controls that optimize multiple objective functions.
    • Please elaborate a reason for the reader to investigate each of the triple cited references [1-3].
    • Please elaborate a reason for the reader to investigate each of the triple cited references [5-7].
    • Please elaborate a reason for the reader to investigate each of the triple cited references [11-13].
    • Please elaborate a reason for the reader to investigate each of the undecuple cited (yikes) references [14-24].
    • Please elaborate a reason for the reader to investigate each of the quadruple cited references [20-23].
  4. Equations are scientifically sound and well presented, enhancing the manuscript quality.
  5. Figures are decently done with some mandatory improvements to ensure the readership has access to the content.
    • Internal font size is legibly sufficiently large.
    • Line styles and sizes are identical in figures 3,8,9,11 rendering the disparate data indistinguishable when the manuscript is read in printed hardcopy (particularly in black and white) negating the value of the figures.
  6. Tables are decently done and aid repeatability of the manuscript.
  7. Please eliminate section 7.

Author Response

Response to Reviewer 1

Interesting research in a very well-prepared manuscript that needs some mild revisions.

Comment 1:

Abstract is okay but is not likely to entice the readership to continue reading the rest of the manuscript.

Use of acronyms/abbreviations in an abstract is unlikely to attract readers not already aware of the manuscript’s content. Consider whether the reader gains anything by seeing in the abstract FCS-MPC, MMPC, DPA, NPC, and CFNPCI.

Results are only presented in weak, qualitative fashion. Highest quality expression of main conclusions or interpretations is quantitative results discussed in the broadest context possible, e.g., percent performance improvement compared to a declared benchmark. “…feasibility and effectiveness are verified…” is very weakly stated results compared to “…xxx percent performance improvement over conventional methods was achieved….”. Seeking quantitative results in the conclusions, the reader only finds “..good steady state performance…”. Meanwhile, line 341 contains quantitative assessment, “…total harmonic distortion of 0.84%....”.

Answer to comment 1:

Thank you very much for reviewer’s comments. The answers to these questions are expressed in two different colors.

Some abbreviations are used in the abstract of the manuscript, which cannot give enough information and attract readers to continue reading. In the Abstract of the revised paper, we try to avoid using those acronyms or abbreviations.

We have improved the expression of the results in the Abstract and the Conclusion to enhance the persuasion of the paper. The last sentence of the Abstract is revised as “Finally, the experimental results based on the designed 9 level prototype show that the develop multilayer model predictive control lead to acceptable steady state, dynamic and robust performance, with only 1.37% of the run time of the TMPC”. The last sentence of the Conclusion is revised as “And the test results show that the TMPC takes 503us to run once, while the MMPC only takes 6.9us to run once. In addition, the experimental results show that the THD of the output voltage is only 0.84% in the steady state experiment, both the load observer and the MMPC could track the step change load and reference within 0.4ms in the dynamic experiment,, and the THD of the output voltage can still reach 3.4% even when the DPA supplies power for diode rectifier bridge load.”.

Action based on comment 1:

In the Abstract of the revised paper, all the acronyms and abbreviations have been removed.

The last sentence of the Abstract has been revised as follow.

Finally, the experimental results based on the designed 9 level prototype show that the develop multilayer model predictive control lead to acceptable steady state, dynamic and robust performance, with only 1.37% of the run time of the TMPC.

The dynamic response time has been retested and added into the last paragraph of Section 5.2.

The THD of the output voltage in the robust experiment has been retested. And the last sentence of the second paragraph of Section 5.3 has been revised as follow.

It can be seen that even when the voltage mode DPA supplies power to a non-linear load, the THD of the output voltage still reaches 3.4%, which shows strong system robustness.

The last sentence of the Abstract has been revised as follow.

And the test results show that the TMPC takes 503us to run once, while the MMPC only takes 6.9us to run once. In addition, the experimental results show that the THD of the output voltage is only 0.84% in the steady state experiment, both the load observer and the MMPC could track the step change load and reference within 0.4ms in the dynamic experiment,, and the THD of the output voltage can still reach 3.4% even when the DPA supplies power for diode rectifier bridge load.

Comment 2:

Please elaborate a reason for the reader to investigate each of the triple cited references [1-3].

Please elaborate a reason for the reader to investigate each of the triple cited references [5-7].

Please elaborate a reason for the reader to investigate each of the triple cited references [11-13].

Please elaborate a reason for the reader to investigate each of the undecuple cited (yikes) references [14-24].

Please elaborate a reason for the reader to investigate each of the quadruple cited references [20-23].

Answer to comment 2:

Thank you very much for reviewer’s comments.

In the Introduction of the revised paper, we have added a reason for the reader to investigate each of Reference [1-3], [5-7], [11-13], [14-24], [20-23].

Action based on comment 2:

The following sentences have been added into the Introduction of the revised paper.

Power amplifiers are widely used in industrial testing and measurement [1]. They are also often used to drive underwater acoustic transducers to produce low-frequency tunable sound sources, which can realize submarine navigation and ranging [2]. Currently power amplifiers can be divided into class A, class B, class AB and class D. Class D power amplifier, also known as digital power amplifier (DPA), is widely used in high-power occasions because of its advantages of low power loss and high efficiency [3].

Compared with two-level or three-level inverter, cascaded inverter, such as cascaded H-bridge inverter [5], modular multilevel converter (MMC) [6], and cascaded full-bridge neutral point clamped inverter (CFNPCI) [7], has the advantages of high power, low step voltage and step current, strong fault handling ability, modular structure and high output waveform quality, thus it is often employed by high power DPAs.

In [14], a MPC of MMC with an independent arm-balancing control is proposed to decouple the circulating current and arm voltage with an arm-balancing control. In [15], a novel MPC algorithm with reduced switching frequency is proposed for MMC. However, the computation of the controller required by the method in [14]-[15] is not reduced.

In [11], a new MPC which selects two voltage levels in a single control period is proposed to reduce the computation burden for cascaded H-bridge inverter.

In [12], the optimization problem of the MPC is solved by the sphere decoding algorithm. However, the sphere decoding algorithm occupies lots of computing resources all the same.

Comment 3:

Equations are scientifically sound and well presented, enhancing the manuscript quality.

Answer to comment 3:

Thank you very much for reviewer’s recognition of our work and the kind suggestions.

We have tried our best to revise this paper. And we look forward to your reviewing our revised paper again.

Comment 4:

Figures are decently done with some mandatory improvements to ensure the readership has access to the content.

Internal font size is legibly sufficiently large.

Line styles and sizes are identical in figures 3,8,9,11 rendering the disparate data indistinguishable when the manuscript is read in printed hardcopy (particularly in black and white) negating the value of the figures.

Answer to comment 4:

Thank you very much for reviewer’s comments. The answers to the two questions are expressed in two different colors.

The font sizes inside all figures have been adjusted to ensure that they are large enough.

Line styles and sizes in Figure 3 have been revised so that they can be distinguishable when the manuscript is read in printed hardcopy (particularly in black and white). We are sorry that Line styles and sizes in Figure 8, 9, and 11 cannot be revised, because they are obtained from Tektronix Oscilloscope, which cannot supply figures with different line styles and sizes.

Comment 5:

Tables are decently done and aid repeatability of the manuscript.

Answer to comment 5:

Thank you very much for reviewer’s recognition of our work and the kind suggestions.

Comment 6:

Please eliminate section 7.

Answer to comment 6:

Thank you very much for reviewer’s comments. We have proofread the manuscript, and Section 7 will not appear in the revised paper.

Reviewer 2 Report

Comments to the authors

Manuscript ID: electronics-1261285

Title: Multilayer Model Predictive Control for the Voltage Mode Digital Power Amplifier Employing Cascaded Full-bridge NPC Inverter

1) State clearly the contribution(s) of the manuscript, and provide sufficient evidences to support your claim.

2) I agree with the authors that MPC is computationally expensive. However, there are some other methods that could be used to address this practical issue. For instance, take a look at Explicit Reference Governor (e.g., 10.1109/LCSYS.2019.2913455) and control barrier functions (e.g., 10.23919/ECC.2019.8796030), and discuss them as alternatives to MPC. I believe that these methods could also work well for the considered problem.

3) It is mentioned that an estimator has been used to estimate the current i_o, instead of a current sensor. Do you consider possible estimation errors in the control system?

4) I failed to place the final optimization problem associated with the MPC.

5) It would be great to compare MPC with the develop multilayer MPC with respect to performance.  Table 4 compares only computational complexity.

Author Response

Response to Reviewer 2

Comment 1:

State clearly the contribution(s) of the manuscript, and provide sufficient evidences to support your claim.

Answer to comment 1:

Thank you very much for reviewer’s comments.

We have stated clearly the contributions of the manuscript in the Abstract, the last paragraph of the Introduction and the Conclusion of the revised paper.

In Section 5, we specifically compared the calculations required for the experimental prototypes using traditional model predictive control and the multilayer model predictive control proposed in this paper. The comparison results show that the calculation amount required by the multilayer model predictive control algorithm is significantly less than that required by the traditional model predictive control, which provide sufficient evidences to support the claim.

Action based on comment 1:

The following sentences have been added to Section 5 of the revised paper.

In the designed experimental prototype, the number of submodules n is equal to 2, and the maximum total output level M can be taken as 4. Based on Table 4, the TMPC requires 1863 addition calculations, 1296 subtraction calculations, 2754 multiplication calculations, 729 division calculations, and 405 comparison calculations for each control period, which poses a huge challenge to the computing performance of the controller. However, the MMPC proposed in this paper only requires 5 addition calculations, 9 subtraction calculations, 5 multiplication calculations, 0 division calculations, and 11 comparison calculations for each control period. In addition, both the TMPC and the MMPC are tested on a DSP of TMS320F28335. The results show that the TMPC takes 503us to run once, while the improved algorithm only takes 6.9us to run once. Therefore, the develop multilayer structure greatly reduces the computation burden of the controller and can be acceptable for general digital controllers.

Comment 2:

It is mentioned that an estimator has been used to estimate the current io, instead of a current sensor. Do you consider possible estimation errors in the control system?

Answer to comment 2:

Thank you very much for reviewer’s comments.

We have considered possible estimation error in the control system. The magnitude of the estimation error is related to the gain parameters of the designed load observer. Reasonable gain parameters can make the estimation error of io very small. Reference [1] provides a better method for selecting gain parameters. In this paper, the selection of load observer gain parameters is based on the method in [1].

[1] H. Nguyen, E. Kim, I. Kim, H. Choi, and J. Jung, “Model predictive control with modulated optimal vector for a three-phase inverter with an LC filter,” IEEE Trans. Power Electron., vol. 33, no. 3, pp. 2690–2703, Mar. 2018.

Comment 3:

I failed to place the final optimization problem associated with the MPC.

Answer to comment 3:

Thank you very much for reviewer’s comments.

Reference [2] shows that the operation mechanism is to solve a finite-time open-loop optimization problem online at each sampling instant, which works based on predicting the system variables and the cost function. The ultimate goal of the final optimization problem is also to obtain the optimal switching state based on predicting the state variables and the cost function.

[2] E. Camacho and C. Bordons, Model Predictive Control. London, U.K.:Springer, 1999.

Comment 4:

It would be great to compare MPC with the develop multilayer MPC with respect to performance. Table 4 compares only computational complexity.

Answer to comment 4:

Thank you very much for reviewer’s comments.

Indeed, the reviewer’s opinion is very good. The performance of the develop multilayer model predictive (MMPC) control should be compared with that of the traditional model predictive control (TMPC). However, the system state variables and the cost function should be repeatedly calculated and compared 81 times in each control period, which requires 1863 addition calculations, 1296 subtraction calculations, 2754 multiplication calculations, 729 division calculations, and 405 comparison calculations for each control period. Therefore, if we apply the TMPC to the designed experimental prototype, the TMPC algorithm may not be able to run to the end within the 25us control period, thus the experimental results when the TMPC is used cannot be provided.

In order to perform fair comparison, we can provide simulation results for comparison. The simulation conditions are exactly the same as the experimental conditions described in Section 5.

The steady state performances of the two control algorithms are firstly compared with each other. Fig. 1 shows the simulation results of the TMPC, where (a) shows the waveforms of the output voltage Vo and its reference Voref, (b) shows the waveforms of the filter inductor current if and its reference ifref, (c) shows the waveforms of the output current io and its estimated value, (d) shows the two capacitor voltages of the first submodule. Fig. 2 shows the simulation results of the develop MMPC.

(a) Vo and Voref                                                                            (b) if and ifref

(c) io estimated value and io measured value                               (d) Capacitor voltages of the first submodule

Fig. 1 Steady state simulation results using TMPC.

(a) Vo and Voref                                                                            (b) if and ifref

(c) io estimated value and io measured value                               (d) Capacitor voltages of the first submodule

Fig. 2 Steady state simulation results using MMPC.

Secondly, the dynamic performances of the TMPC and the MMPC are compared with each other. Fig. 3 shows the simulation results of the TMPC under the condition of the output reference step changing, while Fig. 4 shows the simulation results of the TMPC under the condition of the output reference step changing.

(a) Vo and Voref                                                                            (b) if and ifref

(c) io estimated value and io measured value                               (d) Capacitor voltages of the first submodule

Fig. 3 Dynamic simulation results using TMPC.

(a) Vo and Voref                                                                            (b) if and ifref

(c) io estimated value and io measured value                               (d) Capacitor voltages of the first submodule

Fig. 4 Dynamic simulation results using MMPC.

Thirdly, the robust performances of the TMPC and the MMPC are compared with each other. Fig. 5 shows the robustness simulation results of the TMPC, where (a) shows the waveforms of the output voltage Vo and its reference Voref, (b) shows the waveforms of the filter inductor current if and its reference ifref, (c) shows the waveforms of the output current io and its estimated value, (d) shows the two capacitor voltages of the first submodule. Fig. 6 shows the robustness simulation results of the MMPC.

(a) Vo and Voref                                                                            (b) if and ifref

(c) io estimated value and io measured value                               (d) Capacitor voltages of the first submodule

Fig. 5 Robust simulation results using TMPC.

(a) Vo and Voref                                                                            (b) if and ifref

(c) io estimated value and io measured value                               (d) Capacitor voltages of the first submodule

Fig. 6 Robust simulation results using MMPC.

It can be seen from the above three groups of simulation that the TMPC and the develop MMPC lead to similar steady-state, dynamic and robust control effects. However, the develop MMPC only requires 5 addition calculations, 9 subtraction calculations, 5 multiplication calculations, 0 division calculations, and 11 comparison calculations for each control period. From this aspect, the develop MMPC has obvious advantages.

Action based on comment 5:

The following paragraph have been added to Section 5 in the revised paper.

In the designed experimental prototype, the number of submodules n is equal to 2, and the maximum total output level M can be taken as 4. Based on Table 4, the TMPC requires 1863 addition calculations, 1296 subtraction calculations, 2754 multiplication calculations, 729 division calculations, and 405 comparison calculations for each control period, which poses a huge challenge to the computing performance of the controller. However, the MMPC proposed in this paper only requires 5 addition calculations, 9 subtraction calculations, 5 multiplication calculations, and 11 comparison calculations for each control period. In addition, both the TMPC and the MMPC are tested on a DSP of TMS320F28335. The results show that the TMPC takes 503us to run once, while the improved algorithm only takes 6.9us to run once. Therefore, the develop multilayer structure greatly reduces the computation burden of the controller and can be acceptable for general digital controllers.

Reviewer 3 Report

Please find below some areas of improvement of your paper "Multilayer Model Predictive Control for the Voltage Mode Digital Power Amplifier Employing Cascaded Full-bridge NPC Inverter":

First of all I would like to congratulate you on your work, but I consider the paper requires significant editing. Please make your contribution and the novelty of your work very clear to readers. Authors should add more details and explain how and what methods were taken, analysis should be clear.

1.- The title should be revised. Is too long.

2.- Review the abstract. Please make your contribution and the novelty of your work clear to readers.

3.- Try to avoid too many acronyms. Define acronyms / abbreviations the first time you use them. For example you missed NPC (NPC inverter).

4.- In the abstract, it is not clear what you mean with "seriously restricted".

5.- Review keywords.

6.- You must define all parameters related to equations through the paper. You can use tables. For example: In the introduction: what is (du/dt) ? or  (di/dt),?

7.- There is plenty of literature in this area of research. Add a section for related work / previous work. You must summarize what you have learned, and state clearly the novelty of your work.

8.- If possible, add diagram blocks for a better understanding of the paper.

9.- Review your writing through the whole paper. In most sections the concepts are too general without technical explanation and references. You must make it more precise and to the point. You must understand what you want to address in this paper. Use tables to summarize information.

10.- Authors should add more technical details and explain how and what methods were taken, analysis should be clear.

11.- Review conclusions.

12.- Review the Patent, Appendix A and Appendix B section. 

Author Response

Response to Reviewer 3

Please find below some areas of improvement of your paper "Multilayer Model Predictive Control for the Voltage Mode Digital Power Amplifier Employing Cascaded Full-bridge NPC Inverter":

Comment 1:

First of all I would like to congratulate you on your work, but I consider the paper requires significant editing. Please make your contribution and the novelty of your work very clear to readers. Authors should add more details and explain how and what methods were taken, analysis should be clear.

Answer to comment 1:

Thank you very much for reviewer’s comments.

The main contribution of this paper is that it proposes a multilayer model predictive control for the voltage mode digital power amplifier employing cascaded full-bridge neutral point clamped inverter, which can greatly reduce the amount of calculation without affecting its dynamic performance. We have stated clearly the contributions of the manuscript in the Abstract, the last paragraph of the Introduction and the Conclusion of the revised paper.

Section 3 and Section 4 have been revised. Some details have been added, and some expressions have been modified to make the methods taken clearer.

Action based on comment 1:

The last sentence of the Abstract has been revised as follow.

Finally, the experimental results based on the designed 9 level prototype show that the develop multilayer model predictive control lead to acceptable steady state, dynamic and robust performance, with only 1.37% of the run time of the TMPC.

The last sentence of the conclusion has been revised as follow.

And the test results show that the TMPC takes 503us to run once, while the MMPC only takes 6.9 us to run once. In addition, the experimental results show that the THD of the output voltage is only 0.84% in the steady state experiment, both the load observer and the MMPC could track the step change load and reference within 0.4ms in the dynamic experiment, and the THD of the output voltage can still reach 3.4% even when the DPA supplies power for diode rectifier bridge load.

Comment 2:

The title should be revised. Is too long.

Answer to comment 2:

Thank you very much for reviewer’s comments.

The title of the manuscript has been revised as “Multilayer Model Predictive Control for a Voltage Mode Digital Power Amplifier” in the revised paper.

Action based on comment 2:

The title of the manuscript has been revised as “Multilayer Model Predictive Control for a Voltage Mode Digital Power Amplifier” in the revised paper.

Comment 3:

Review the abstract. Please make your contribution and the novelty of your work clear to readers.

Answer to comment 3:

Thank you very much for reviewer’s comments.

We have revised the Abstract of the manuscript, and the contribution and the novelty of this paper has been emphasized in the revised Abstract.

Action based on comment 3:

The last sentence of the Abstract has been revised as follow.

Finally, the experimental results based on the designed 9 level prototype show that the develop multilayer model predictive control lead to acceptable steady state, dynamic and robust performance, with only 1.37% of the run time of the TMPC.

Comment 4:

Try to avoid too many acronyms. Define acronyms / abbreviations the first time you use them. For example you missed NPC (NPC inverter).

Answer to comment 4:

Thank you very much for reviewer’s comments.

We have deleted some unnecessary acronyms in the revised paper. And all the acronyms/ abbreviations have been defined when they are used for the first time. For example, the acronym NPC has been defined in the third paragraph of Section 1 of the revised paper.

Comment 5:

In the abstract, it is not clear what you mean with "seriously restricted".

Answer to comment 5:

Thank you very much for reviewer’s comments.

We are sorry that we used inaccurate English expressions in the manuscript. In the revised paper, we have replaced “seriously restricted” with “severely limited”, which means that the computational complexity of the finite control set model predictive control limits its application to cascaded inverters.

Action based on comment 5:

In the Abstract of the revised paper, “seriously restricted” has been revised as “severely limited”.

In the third paragraph of Section 1 of the revised paper, “seriously restricts” has been revised as “severely limits”.

Comment 6:

Review keywords.

Answer to comment 6:

Thank you very much for reviewer’s comments.

We have reviewed the keywords of this paper, and the acronyms/abbreviations have been removed in the revised paper.

Action based on comment 6:

The keywords of the manuscript have been revised as follow.

Model predictive control; Cascaded full-bridge neutral point clamped inverter; Load observer; Calculation Amount

Comment 7:

You must define all parameters related to equations through the paper. You can use tables. For example: In the introduction: what is (du/dt)? or (di/dt),?

Answer to comment 7:

Thank you very much for reviewer’s comments.

We have checked all the parameters related to equations through this paper, and ensure that all parameters are defined. The term, (du/dt), denotes the step voltage, (di/dt) denotes the step current. In the revised paper, in order to avoid misunderstanding of (du/dt) and (di/dt), we have removed (du/dt) and (di/dt) in the introduction of the revised paper.

Comment 8:

There is plenty of literature in this area of research. Add a section for related work / previous work. You must summarize what you have learned, and state clearly the novelty of your work.

Answer to comment 8:

Thank you very much for reviewer’s comments.

There are indeed many literatures published in this area of research. We have summarized what we learned from the related work or previous work in the second and third paragraph of the Introduction of the revised paper, and emphasized the novelty of our work in the fourth paragraph of the Introduction of the revised paper.

Comment 9:

If possible, add diagram blocks for a better understanding of the paper.

Answer to comment 9:

Thank you very much for reviewer’s comments.

We have added a diagram blocks in Section 4 of the revised paper.

Comment 10:

Review your writing through the whole paper. In most sections the concepts are too general without technical explanation and references. You must make it more precise and to the point. You must understand what you want to address in this paper. Use tables to summarize information.

Answer to comment 10:

Thank you very much for reviewer’s comments.

We have checked the paper to eliminate grammatical errors. Some concepts have been explained technically under the support of references to make them more precise and to the point. We have also revised the full paper to clearly show what we want to address in this paper. Table 5 is added to summarized the experimental parameters.

Comment 11:

Review conclusions.

Answer to comment 11:

Thank you very much for reviewer’s comments.

The conclusion of this paper has been reviewed and revised to show the contribution of this paper clearer.

Action based on comment 11:

The last sentence of the conclusion has been revised as follow.

And the test results show that the TMPC takes 503us to run once, while the MMPC only takes 6.9us to run once. In addition, the experimental results show that the THD of the output voltage is only 0.84% in the steady state experiment, both the load observer and the MMPC could track the step change load and reference within 0.4ms in the dynamic experiment,, and the THD of the output voltage can still reach 3.4% even when the DPA supplies power for diode rectifier bridge load.

Round 2

Reviewer 2 Report

No further comments. 

Author Response

Thanks very much for reviewer’s comments.

We have tried our best to revise this paper. And we look forward to your reviewing our revised paper again.

Reviewer 3 Report

Please find below some areas of improvement of your paper "Multilayer Model Predictive Control for a Voltage Mode Digital Power Amplifier"
Congrats again on your work. I encourage you to submit a revised final version of your work.

1.- In the paper you must include some comments or analysis of propagation of errors.
How did you take into account the different types of errors? How did they impact the results?

And again some general recommendations:

2.- Avoid acronyms in the abstract. For example: TMPC

Define acronyms / abbreviations the first time you use them.

3.- The abstract can be improved. 
Review the abstract. Please make your contribution and the novelty of your work clear to readers.

4.- Avoid too many acronyms.

5.- Define all the parameters related to equations through the paper. I recommend to use tables.

Author Response

Response to Reviewer 3

Please find below some areas of improvement of your paper "Multilayer Model Predictive Control for a Voltage Mode Digital Power Amplifier".

Congrats again on your work. I encourage you to submit a revised final version of your work.

Comment 1:

In the paper you must include some comments or analysis of propagation of errors. How did you take into account the different types of errors? How did they impact the results?

Answer to comment 1:

Thank you very much for reviewer’s comments. The answers to these questions have been shown in different colors.

In the revised paper, the comments or analysis of both the estimating error and the tracking error have been supplemented. The analysis of the estimating error of the designed load observer is supplemented into Section 3. The analysis of the estimating error of the designed load observer is supplemented into Section 3. The analysis of the tracking error of the proposed MMPC is supplemented into Section 4.1.

The estimating error of the designed load observer is related to the selected gain matrix K. We have selected a reasonable gain matrix K to get a small estimation error. The tracking error is not only related to the estimating error, but also related to the control period. A smaller estimating error and a smaller control period will lead to a smaller tracking error.

The impacts of the errors on the results have been supplemented into Section 5.1 and Section 5.2 of the revised paper.

Action based on comment 1:

The following paragraph has been added into Section 3 of the revised paper.

It should be pointed out that the estimation error of the designed load observer depends on the selected gain matrix K. Unreasonable gain parameters will lead to large estimation error of io, and eventually lead to tracking error of the MMPC in Section 4. Therefore, the gain matrix of the load observer should be elaborated to achieve a tradeoff between the dynamic performance and the noise immunity. The parameter selection method in [26] can be used to determine the gain matrix K.

The following paragraph has been added into Section 4.1 of the revised paper.

Based on the above analysis, the tracking error of the MMPC is not only related to the estimation accuracy of the output current io, but also related to its control period. A shorter control period will lead to a smaller tracking error. The upper layer control significantly reduces the computation and the running time of the MPC algorithm, so that a shorter control period can be used and a smaller tracking error can be obtained.

The first sentence of the second paragraph of Section 5.1 has been revised as follow.

It can be seen from Fig. 8 that the designed load observer accurately estimates the actual load current io under steady state condition. The small estimating error of io leads the proposed MMPC to output accuracy inductor current and output voltage, which means small tracking errors are obtained.

The first sentence of the second paragraph of Section 5.2 has been revised as follow.

It can be seen from Fig. 9 that, within 0.4ms, the designed load observer can still quickly and accurately estimate the actual load current under the condition of both the output voltage reference and the load current step change. This also means that the selected gain parameters of the load observer have a good tradeoff between the dynamic performance and the noise immunity. And with the help of this, the proposed MMPC also quickly and accurately tracks the step change output reference.

The following reference has been added in the revised paper.

[26] H. Nguyen, E. Kim, I. Kim, H. Choi, and J. Jung, “Model predictive control with modulated optimal vector for a three-phase inverter with an LC filter,” IEEE Trans. Power Electron., vol. 33, no. 3, pp. 2690–2703, Mar. 2018.

Comment 2:

Avoid acronyms in the abstract. For example: TMPC. Define acronyms / abbreviations the first time you use them.

Answer to comment 2:

Thank you very much for reviewer’s comments.

In the revised paper, we have deleted the acronym, TMPC, in the Abstract. And we have also checked all the paper to ensure all the acronyms/abbreviations are defined the first time they are used.

Comment 3:

Review the abstract. Please make your contribution and the novelty of your work clear to readers.

Answer to comment 3:

Thank you very much for reviewer’s comments.

We have revised the Abstract of the manuscript, and the contribution and the novelty of this paper has been shown to readers in the following sentence.

In this paper, a load observer based multilayer model predictive control is proposed for the voltage mode digital power amplifier employing cascaded full-bridge neutral point clamped inverter, which can avoid the use of load current sensor and greatly reduce the controller computation without affecting its dynamic performance.

Action based on comment 3:

The following sentence has been added into the Abstract of the revised paper.

In this paper, a load observer based multilayer model predictive control is proposed for the voltage mode digital power amplifier employing cascaded full-bridge neutral point clamped inverter, which can avoid the use of load current sensor and greatly reduce the controller computation without affecting its dynamic performance.

Comment 4:

Avoid too many acronyms.

Answer to comment 4:

Thank you very much for reviewer’s comments.

In the revised paper, some acronyms such as DPA, MMC, THD, FCS-MPC, TFCS-MPC have been deleted.

Comment 5:

Define all the parameters related to equations through the paper. I recommend to use tables.

Answer to comment 5:

Thank you very much for reviewer’s comments.

According to the reviewer's suggestion, we have added a table to the revised paper to define all parameters related to equations.

Action based on comment 5:

The following contents have been added into the revised paper.

Appendix

All the parameters related to equations are defined in the following table.

UCi1

Capacitor voltage of Ci1

Estimated value of Y

UCi2

Capacitor voltage of Ci2

Estimated value of if

Vdc

Voltage of Submodule dc power supply

Estimated value of Vo

Lf

Filter inductor

Estimated value of io

Cf

Filter capacitor

K

Gain matrix of the load observer

if

Current of the if

Voref

Control reference of Vo

Vo

Voltage of Cf

Ifref

Control reference of if

Vab

Output voltage of the CFNPCI

J(h)

Total cost function

Mi

Output level of Submodule i

J1(h)

Cost function for filter inductor current control

M

Total output level

J2(h)

Cost function for output voltage control

x

System state variable

h1

Solution of J1(h)=0

io

Output current

h2

Solution of J2(h)=0

X

Extended state variable

λ1

Weight factor for filter inductor current control

Y

Output variable of the load observer

λ2

Weight factor for output voltage control

Estimated value of X

TS

Control period
